Cerrado–Caatinga; flagship species; habitat loss; national action plan; *Tolypeutes*

**Author for correspondence:**
Anderson Feijó,
Email: andefeijo@gmail.com

# Defining priority areas for conservation of poorly known species: A case study of the endemic Brazilian three-banded armadillo

Anderson Feijó[1] (ORCID), Rodolfo Magalhães[2,3,4] (ORCID), Adriana Bocchiglieri[5],
José Luís P. Cordeiro[6,7,8], Liana Mara Sena[2,3] and Nina Attias[9,10] (ORCID)

[1]Key Laboratory of Zoological Systematics and Evolution, Institute of Zoology, Chinese Academy of Sciences, Beijing, China; [2]Programa de Pós-Graduação em Ecologia, Conservação e Manejo da Vida Silvestre, Universidade Federal de Minas Gerais, Belo Horizonte, Brazil; [3]Departamento de Genética, Ecologia e Evolução, Universidade Federal de Minas Gerais, Belo Horizonte, Brazil; [4]EDGE of Existence Programme, Zoological Society of London, London, UK; [5]Programa de Pós graduação em Ecologia e Conservação, Universidade Federal de Sergipe, São Cristóvão, Brazil; [6]Fundação Oswaldo Cruz, Fiocruz Ceará, Rio de Janeiro, Brazil; [7]International Platform for Science, Technology, and Innovation in Health, Aveiro, Portugal; [8]Department of Biology and Centre for Environmental and Marine Studies, Aveiro University, Aveiro, Portugal; [9]Department of Wildlife Ecology and Conservation, University of Florida, Gainesville, FL, USA and [10]ICAS – Instituto de Conservação de Animais Silvestres, Campo Grande, Brazil

**Abstract**

Conservation of poorly known species is challenging as lack of knowledge on their specific requirements may hamper effective strategies. Here, by integrating biogeographical and landscape analyses, we show that informed actions can be delineated for species with limited presence-only data available. We combine species distribution and connectivity models with temporal land cover changes to define priority areas for conservation of the endemic Brazilian three-banded armadillo, one of the most threatened xenarthrans that was once considered extinct in the wild. We revealed that areas of savanna and grassland are the most suitable habitats for the species and that uplands in the Caatinga ecoregion have a greater likelihood for dispersal. The few remnant armadillo populations are spatially associated with core areas of natural vegetation remnants. Worrisomely, 76% of natural core areas were lost in the past 30 years, mirroring the species' severe population decline. Preserving the remnant core natural areas should be a high priority to ensure the species' survival. We highlight key areas for proactive and reactive conservation actions for the three-banded armadillo that will benefit other threatened sympatric species. Our integrative framework provides a set of valuable information for guided conservation management that can be replicated for other poorly known species.

**Impact statement**

Here, we combine a set of methodological approaches to define priority areas for proactive and reactive conservation management of poorly known species. We used the Brazilian endemic three-banded armadillo as a case study, a species once thought to be extinct in the wild and currently known by a few dozen records in northeastern Brazil. Our framework integrating species distribution and connectivity models revealed key habitats and ecological conditions dictating the species distribution and identified main corridors for connecting the remaining populations. Assessing habitat loss across the species range using high-resolution historical land cover data, we found that nearly 6 million hectares of natural vegetation were converted to pasture and agriculture in the past 35 years. In addition, we show that the few known remnant populations of the three-banded armadillo are closely associated with remnant core areas of savanna and grassland, while historical records, reflecting populations now extirpated, were concentrated in areas facing severe habitat loss. Therefore, well-preserved natural vegetation remnants are key to ensuring the species' long-term survival. Based on our multiple findings, we highlight priority areas for resource allocation and conservation actions that can restore connectivity and ultimately benefit other open-dweller species. Our study shows how informed conservation actions can be delineated for threatened species with only limited presence-data records available, a method that can be readily replicated for other poorly known species.



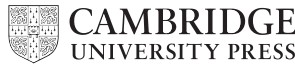

## Introduction

Well-planned conservation strategies require solid knowledge of species requirements and associated threats. Such information is nevertheless limited for most species and nonexistent

for most rare taxa. Hence, strategies to better guide research and conservation endeavors for poorly known species are necessary, even in light of very limited data (Marcer et al., 2013).

Modern analytical tools have been widely applied in conservation science to optimize resource allocation and define prioritization schemes. Biogeographical analyses are particularly powerful in guiding spatial conservation planning (Whittaker et al., 2005; Franklin, 2013; Villero et al., 2016) and their results can be easily interpretable by multiple stakeholders (Ferraz et al., 2021a). For example, species distribution modeling and landscape connectivity modeling have been extensively used to direct field surveys, reveal key environmental factors shaping species distribution, and highlight paths to build habitat corridors and protected areas (PAs; Williams et al., 2009; Dickson et al., 2013; Villero et al., 2016; Bonnin et al., 2020; Ferraz et al., 2021b). These methods can be critically applied using raw occurrence (presence-only) data, often the only available information for poorly known species. Nevertheless, additional data can be integrated to increase model performance (Franklin, 2013; Villero et al., 2016; Baker et al., 2020; Frans et al., 2021) and the results can be later refined when new studies become available (Marcer et al., 2013; McShea, 2014).

These tools are further useful to identify potential threats spatially and highlight important areas for conservation actions (Brooks et al., 2006; Romero-Muñoz et al., 2019; Doré et al., 2021; Swan et al., 2021; Feijó et al., 2022). One of the most assessed features when setting conservation priorities is the degree of vulnerability (Brooks et al., 2006). Areas facing high alteration rates often require immediate intervention (i.e., reactive action), while areas deemed as of high biological value but facing low threat can be managed proactively by applying measures that will prevent or reduce future impacts. Although the current threats to biodiversity are multifactorial, habitat loss is indisputably the leading factor (Brooks et al., 2002). Anthropogenic land modification can thus be used as a proxy of biodiversity vulnerability and integrated with biogeographical models to define areas demanding different management regimes.

Among mammals, xenarthrans (armadillos, sloths, and anteaters) are one of the four main lineages of eutherians and the only one that originated in South America (Gibb et al., 2016). They display exclusive physiological and anatomical features and are represented by a few dozen modern species, which makes them highly irreplaceable (Aguiar and Fonseca, 2008; Superina and Loughry, 2015). Of the 39 living xenarthrans, the Brazilian three-banded armadillo (hereafter three-banded armadillo, *Tolypeutes tricinctus*) is the only armadillo endemic to Brazil and one of the most threatened xenarthrans in the world. It has been classified as threatened since the early 1970s (Coimbra-Filho, 1972) and was listed as 'extinct or probably extinct' in the wild in the 1990s (Cole et al., 1994). Since then, incidental reports revealed the existence of 11 remaining populations (e.g., Santos et al., 1994; Marinho-Filho et al., 1997; Bocchiglieri et al., 2010; Feijó et al., 2015; Magalhães et al., 2021). Currently, the species is classified as vulnerable by the IUCN Red List (Miranda et al., 2014) and as endangered by the Brazilian list of threatened fauna (Reis et al., 2015). Information on the status of the remnant populations is virtually nonexistent, rendering this species a priority for research and conservation (Superina et al., 2014; Feijó et al., 2022).

The current rarity of the three-banded armadillo seems a direct outcome of anthropogenic activities. Habitat loss and hunting are considered the prime causes of its population decline with current estimations predicting a reduction of at least 50% in the last 30 years (Miranda et al., 2014; Reis et al., 2015). To reverse this prospect of extinction, in 2014 the Brazilian environmental agency (ICMBio), following the Convention on Biological Diversity's provisions, sponsored the development of a national action plan for the three-banded armadillo conservation (*Plano de Ação Nacional para a Conservação do Tatu-Bola*, PAN Tatu-Bola) with six priority objectives and 38 associated actions defined in agreement by scientists, NGOs, and other stakeholders. These objectives seek to improve knowledge on the species biology, quantify and assess the status of remaining populations, evaluate main threats across species range, delineate priority areas for conservation, and implement management programs (ICMBio, 2014).

Motivated by these objectives, here, we combine biogeographical and landscape analyses to fill part of the knowledge shortfalls and identify priority areas for informed conservation practices for the poorly known three-banded armadillo. We used species distribution modeling to estimate suitable habitats and reveal environmental factors dictating its distribution. We further quantify the main land cover classes and level of protection across the species range. In addition, we applied circuit theory to identify key corridors for maintaining connectivity among the few known remnant armadillo populations. The mapping products derived from the above analyses were integrated with assessments of habitat loss in the past 35 years to highlight regions for proactive and reactive conservation actions.

## Materials and methods

### Study region

The three-banded armadillo is restricted to the Caatinga and adjacent Cerrado savannas of Brazil (Figure 1; Feijó et al., 2015). The Caatinga is an ecoregion exclusive to northeastern Brazil and is South America's largest seasonally dry forest, covering an area of 912,529 km$^2$ (Silva et al., 2017). The region has a hot, semi-arid climate with marked seasonal rainfall and average annual precipitation that ranges from 200 to 800 mm. Caatinga elevation ranges from 0 to 1,877 m a.s.l., with an average of 406 m. Recent estimation shows that 63% of the Caatinga has been converted into anthropogenic ecosystems (Silva and Barbosa, 2017) and the main causes of habitat loss are slash-and-burn agriculture, overgrazing by livestock, and firewood harvesting (Leal et al., 2005). The Cerrado is a vast tropical savanna of almost 2 million km$^2$. The region is a global biodiversity hotspot that presents a semi-humid tropical climate with marked dry and rainy seasons, annual precipitation that ranges from 800 to 2,000 mm, and an average elevation of 480 m a.s.l. Native vegetation loss in the Cerrado is estimated around 46% mainly caused by monocultures and pastures (Strassburg et al., 2017; Vieira et al., 2019).

### Species occurrence data

We compiled occurrence records of three-banded armadillos from literature, public databases, and field records (Supplementary Table S1). To maximize the number of field records, we contacted researchers working within the known distribution of the species to inquire about the occurrence of the three-banded armadillo in their study sites. The reliability of these records was confirmed based on photographs and detailed descriptions provided by the researchers. Because one of our main goals is to propose corridors to improve connectivity among remnant populations, we focused our modeling analyses on recent records (since the year 2000) as historical records might represent populations now extirpated (Feijó et al.,

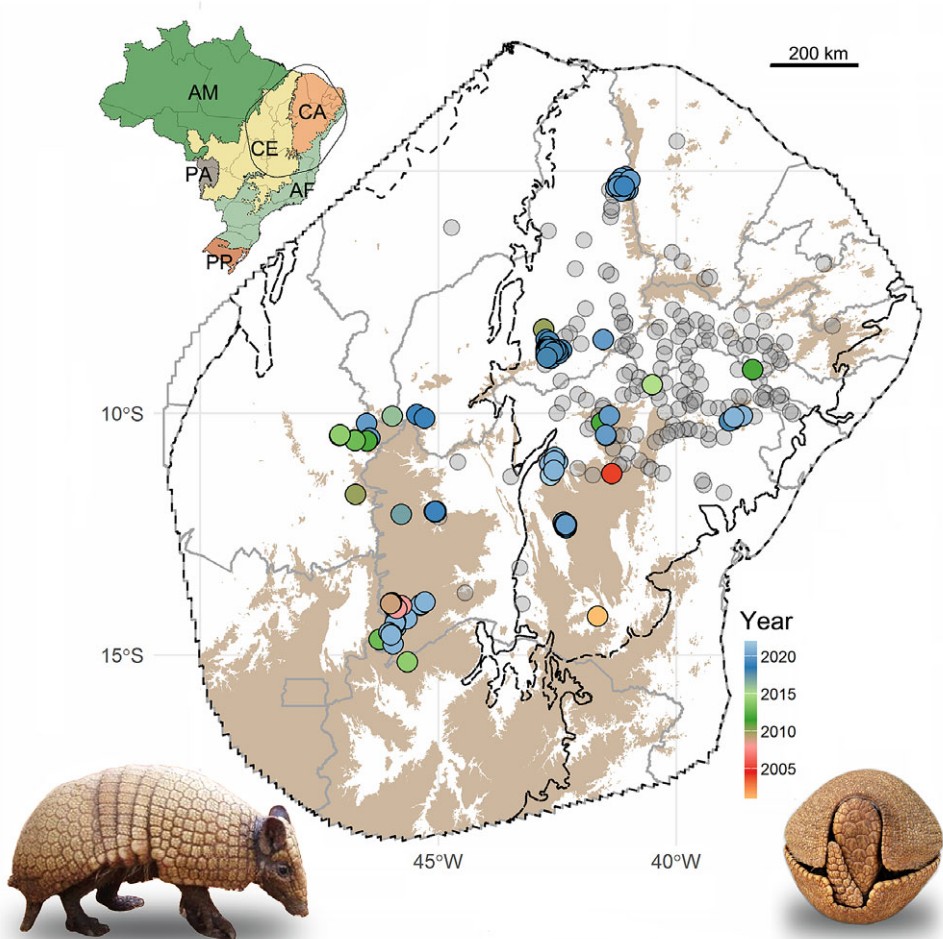

**Figure 1.** Localities recorded for the Brazilian three-banded armadillo in northeastern Brazil. *Note:* Recent records (since the year 2000) are color-coded by the year of the record. Historical records (pre-2000) are in gray. Upland (>625 m) areas are shown in brown. Black lines delimit the biomes (according to Instituto Brasileiro de Geografia e Estatística (IBGE), 2004) and gray lines delimit the Brazilian states. Inset map shows Brazilian biomes and the limits of the convex hull set at 400 km from the marginal recent records. AM, Amazon; CA, Caatinga; CE, Cerrado; AF, Atlantic Forest; PA, Pantanal; PP, Pampas.

2015). Moreover, most of the historical records lack accurate geographic coordinates, and using the coordinates of nearby towns as proxies may bias our analyses and would require reducing the resolution of the models (Baker et al., 2020).

### Habitat suitability modeling

To predict areas with high habitat suitability for three-banded armadillo, we performed species distribution models (SDMs) using the ENMTML R package (Andrade et al., 2020). To reduce sampling bias and spatial autocorrelation, duplicated localities within a 1 km radius were discarded. Because random background points (BPs) can produce biased predictions (Phillips et al., 2009), we allocated BP outside a buffer area of 4 km around each known locality and based on the lowest suitable region predicted by a Bioclim model (Lobo et al., 2010; Andrade et al., 2020). The rationale of this approach is to use the known occurrence points to better guide BP placement, taking into account both geographic and environmental space, which has been shown to improve model performance (Phillips et al., 2009; Lobo et al., 2010). The ratio between presence and BPs was 1:1. The environmental predictors were selected as the least correlated (Pearson $r < 0.7$) from a set of

22 layers describing topography (elevation and roughness), climatic conditions, and vegetation. Climatic parameters include the 19 layers obtained from WorldClim 2 (Fick and Hijmans, 2017) and vegetation is represented by the mean annual leaf area index (LAI) derived from monthly estimates retrieved for the year 2020 (Myneni and Knyazikhin, 2018). All environmental layers have a 30-s arc resolution.

Maps of habitat suitability were produced based on five modeling algorithms: generalized linear models, generalized additive models, boosted regression trees, random forest, and maxent. The performance of our models was tested by partitioning the localities into testing and training bins using the 'k-fold' method with five folds. Models were evaluated based on Area Under the Curve, True Skill Statistics (TSS), and omission rate parameters. To increase predictive performance and account for uncertainty associated with each modeling algorithm, we created an ensemble output based on the average of suitability values weighted by the algorithm-specific TSS performance (Andrade et al., 2020; Hao et al., 2020). Finally, to provide a better approximate estimation of the species' occupied areas while avoiding potential but inaccessible regions (Peterson and Soberón, 2012), we masked the ensemble output with a 400 km buffered convex hull surrounding the recent occurrence records.

## Associated environmental parameters

To better visualize and define the environmental parameters closely associated with three-banded armadillo distribution, we first classified the probabilistic SDM ensemble output into six habitat suitability categories: unsuitable (including grid cells with suitability values below the 10th percentile), very low (between 10th and 30th percentiles), low (30th–50th percentiles), moderate (50th–70th percentiles), high (70th–90th percentiles), and very high (above 90th percentile). We then extracted the values of the 22 environmental variables from each cell using the raster R package (Hijmans, 2021) and explored the environmental segregation between suitability classes using principal component analysis (PCA). Here, we aimed to determine which set of variables were most closely associated with areas of high and very high habitat suitability. All environmental data were standardized prior to analysis.

We further explored land-cover types present across the habitat suitability classes. We used the 2020 land-cover layer from MapBiomas collection 6 (MapBiomas, 2020) and resampled it from the original resolution (~30 m) to a 30-s arc resolution (~1 km) to match the other environmental layers. We combined all types of agricultural land uses (e.g., soybean, sugarcane, and coffee) into one broad 'agriculture' class (Supplementary Table S2). In addition, we quantified the percentage of the area of each suitability category that is within PAs boundaries. For PA's limits, we used the World Database on Protected Areas (UNEP-WCMC and IUCN, 2019), which includes national, subnational, private, and indigenous PAs.

## Connectivity modeling

To identify areas that, if preserved, may serve as corridors and enable connectivity among three-banded armadillo remnant populations, we applied the circuit theory using Circuitscape 5 (Anantharaman et al., 2020). Briefly, circuit-based analysis quantifies the flow of movement among nodes (connective elements) given the voltage applied and the resistance present in a circuit. In ecology, nodes are usually represented by populations, habitat patches, and PAs, and resistance is represented by metrics that estimate the inverse likelihood of movement of an organism across habitats present in a landscape (McRae et al., 2008). Thus, habitats (e.g., cells in a raster grid) with less resistance will have a higher likelihood of movement (dispersal). In this study, we used recent occurrence records of the species as the connective elements (focal nodes) and due to computational limitations duplicated localities within an 8 km radius were filtered out and treated as one node, resulting in 73 final focal nodes. We used a negative exponential transformation of continuous suitability values from the SDM ensemble output as a proxy of resistance values following Bonnin et al. (2020). Thus, cells with low habitat suitability values have high resistance to movement. Additionally, cells classified as unsuitable, very low, and low habitat suitability in the ensemble model were assigned to infinite resistance, representing complete barriers to dispersal. Therefore, the final resistance grid reflects the likelihood of movement associated with areas of moderate to very high habitat suitability.

Connectivity modeling was based on pairwise comparisons in which effective resistances are derived between all pairs of focal nodes. The resulting cumulative current map represents the sum of individual values obtained from each node pair combination (Anantharaman et al., 2020). To highlight the main paths for connecting three-banded armadillo populations, we focus on areas with current flow values above the 70th percentile (main corridors), reflecting a greater likelihood of movement.

## Areas for proactive and reactive conservation practices

Because habitat loss is considered one of the main threats to three-banded armadillos (Miranda et al., 2014; Reis et al., 2015), we explored spatiotemporal changes in land cover across its distribution. Specifically, we retrieved the land-cover information for the years 1985, 2000, and 2020 from MapBiomas (2020) and masked them with a 200 km buffered convex hull surrounding the recent occurrence records. This buffer comprises most of the regions classified as of high and very high habitat suitability for the species. To simplify visualization and data analyses, we reclassified the original land-cover classes into seven classes: the three most common natural classes (savanna, grassland, and forest), three anthropogenic classes (pasture, agriculture, and mosaic of agriculture and pasture), and 'others'. The latter includes mangroves, wetlands, urban areas, dunes, mining, and forest plantations, covering a very small portion of the region (see 'Results' section).

To assess the temporal variation in land cover composition, we calculated the number and area of patches and the number and area of the core region for each land-cover class for each period using the landscapemetrics R package (Hesselbarth et al., 2019). Patches are defined as contiguous cells belonging to the same land-cover class and core region represents the contiguous cells that are at least two cells (i.e., >2 km) away from the edge of the patch (Hesselbarth et al., 2019). Then, we explored the spatiotemporal changes of these metrics in savannas and grasslands (natural habitats suitable for the species) between 1985 and 2020. Our goal was to identify areas converted to anthropogenic land uses and remnants of core natural areas (refuges), which may help guide reactive and proactive conservation actions, respectively.

To evaluate whether recent populations are located in areas with a greater proportion of natural habitats, we further quantified the proportion of savanna (patch and core) remnants surrounding recent three-banded armadillo records and compared it with that surrounding historical (before the year 2000) armadillo records (Figure 1). To reduce spatial autocorrelation, we discarded duplicated localities within a ~18 km radius and created a squared buffer of 5 km radius surrounding each locality. Using the landscapemetrics R package, we calculated the total sum of patch area, total sum of core area, the average patch area, and the average core area present in each buffer and tested the difference between historical and recent landscapes using a *t*-test.

Finally, to identify important target regions for reactive and proactive conservation managements, we overlapped the SDM ensemble and connectivity maps with that of extant PAs and built kernel density estimation heat maps highlighting areas holding a greater density of core natural remnants.

## Results

### Recent populations

We gathered a total of 151 recent occurrence records for the three-banded armadillo, representing 31 independent localities (i.e., populations; Figure 1 and Supplementary Table S1). Of these, about 78% were recorded in the past 5 years. These records are mainly distributed in the uplands (elevation above 625 m) of the northeastern portion of the Cerrado and the western portion of the Caatinga (Figure 1). Only 24% of the recent records were within

PAs, indicating that most of the known remnant populations inhabit unprotected lands. On the other hand, historical records are mainly located in the lowland central portion of Caatinga (Figure 1).

### Habitat suitability and associated parameters

The SDM ensemble output reveals the northeastern portion of the Cerrado and central portion of the Caatinga as the highest suitability areas for the three-banded armadillo (Figure 2a and see Supplementary File for more details on model performance). Peripheral areas close to the Atlantic Forest on the east and to Amazon on the west and north are deemed unsuitable or of very low suitability. The PCA plot shows that areas of high and very high suitability clustered together in the environmental space, displaying mostly positive values on the second principal axis (Figure 2b). We found that the three-banded armadillo is mainly associated with areas with pronounced annual and diurnal temperature ranges and marked precipitation seasonality (Figure 2c). Moreover, we found that savannas (52.4%), pastures (16.1%), and grasslands (11.7%) currently cover most of the area deemed as of high and very high habitat suitability for the species (Supplementary Table S3). PAs cover up to 21% of the region of greater suitability (Supplementary Table S4).

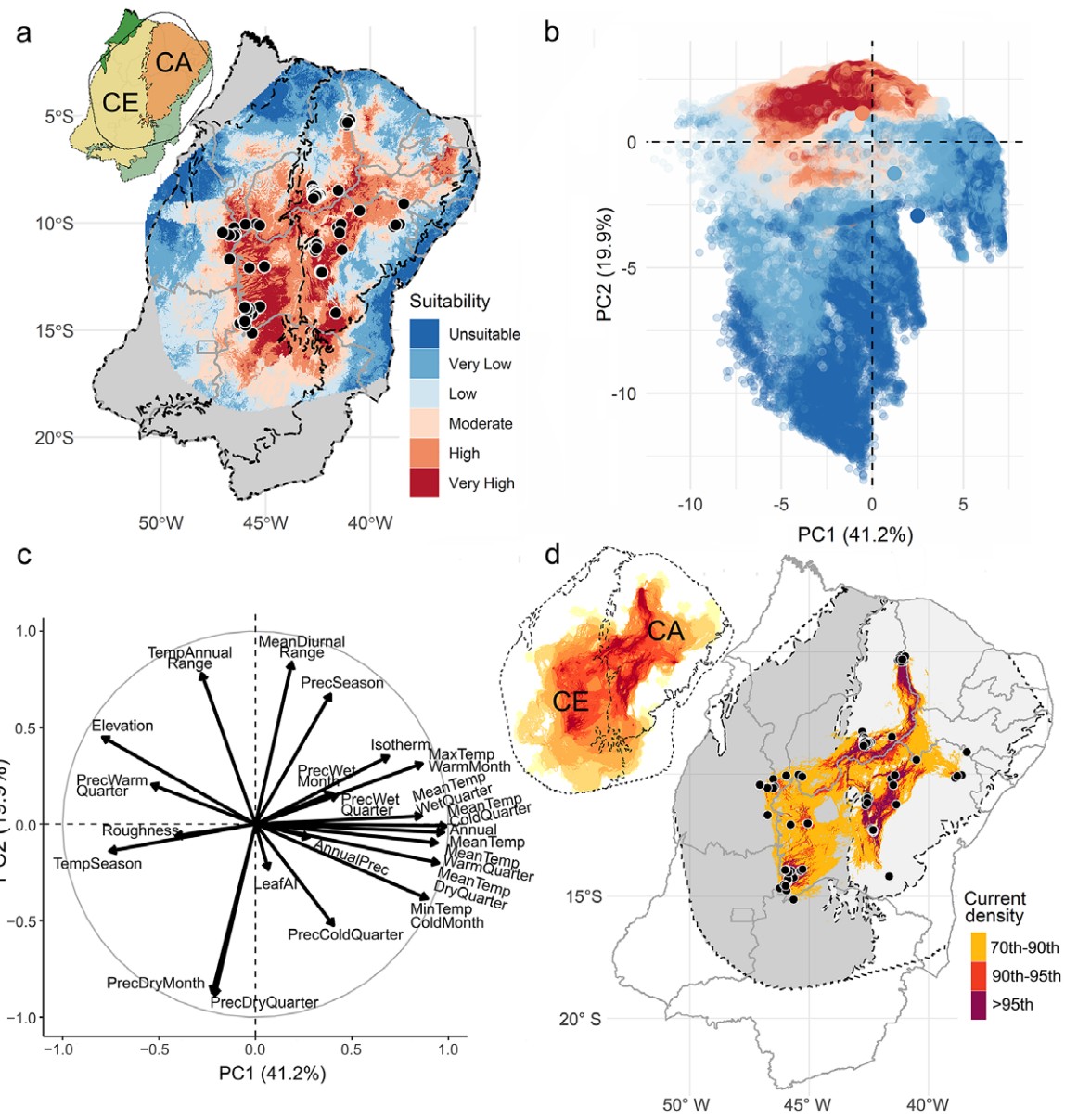

**Figure 2.** Habitat suitability, connectivity, and environmental parameters associated with the distribution of the Brazilian three-banded armadillo. *Note:* (a) Species distribution model (SDM) ensemble output. Colors represent habitat suitability classes (see Methods). Black dots represent recent records (after the year 2000). Inset map shows the biomes in northeastern Brazil (according to Instituto Brasileiro de Geografia e Estatística (IBGE), 2004) and the limit of the convex hull set at 400 km from the marginal recent records. CA, Caatinga; CE, Cerrado. (b) Scatterplot of first and second principal components of the environmental parameters extracted from the SDM ensemble range, showing the environmental segregation among habitat suitability classes. Dots represent grid cells from the SDM ensemble map and are color-coded by habitat suitability classes. (c) Loading plot of the 22 environmental variables showing the correlation of the environmental parameters. (d) Connectivity map based on cumulative current density derived from Circuitscape. Main map shows values above 70th percentile highlighting the main paths for connecting Brazilian three-banded armadillo populations (black dots). Light gray polygon represents CA and dark gray polygon represents CE. Inset map shows the continuous density values.

### Habitat corridors

The cumulative current density map reveals the uplands as important corridors connecting the known remnant populations of three-banded armadillo (Figure 2d). Our models also show that lowland areas in the southwestern portion of the species range are important to facilitate the exchange of individuals between Cerrado and Caatinga populations.

### Areas for reactive and proactive actions

Our temporal land-cover analyses reveal a marked loss of natural lands and expansion of anthropogenic lands in the last 35 years across the distribution of the three-banded armadillo (Table 1 and Figure 3a,b). Grasslands and savannas lost 41% and 35% of core areas, respectively, representing nearly 6 million hectares (Table 1). In contrast, agricultural lands expanded by 14,083% in the patch area and 37,113% in the core area. This scenario is clear when analyzing the spatial distribution of savanna and grassland remnants (Figure 3c,d). Savanna core areas are now mainly restricted to the border between Caatinga and Cerrado and in some isolated areas of central and northern Caatinga. The few grassland core remnants are located inside PAs in the Cerrado (Figure 3d).

Interestingly, we detected a marked spatial overlap between the location of recent occurrence records and the savanna core areas. Indeed, when comparing the landscape surrounding historical and recent records, we found a significant difference across all savanna patch metrics (Supplementary Table S5). Recent records are located in areas holding a greater amount of savanna patches and core areas, while historical records are mainly surrounded by anthropogenic lands, presenting few core natural areas (Supplementary Figure S1).

By integrating the maps of the final outputs of the SDM and connectivity models with those of PAs and savanna core areas, we found that the center and northern Caatinga as well as the transition of the Caatinga–Cerrado are key regions for implementing habitat protection measures. The implementation of protection measures in these regions would ensure the preservation of remnant savanna cores (Figure 3e,f). We also found that 84% of the natural core remnants are located outside of the extant PAs (Figure 4). On the other hand, armadillo populations in the extreme south and east of Caatinga and west of Cerrado demand reactive actions to avoid local extinction given that they are located in unprotected areas under high anthropogenic pressure. To assist project planning at a local level, we provided a map showing which municipalities have a higher relative proportion of savanna core areas (see Supplementary Figure S2).

### Discussion

Prior to this study, recent records of the Brazilian three-banded armadillo were limited to only 11 localities (Feijó et al., 2015; Attias et al., 2016; Campos et al., 2019; Santos et al., 2019; Magalhães et al., 2021). In contrast, local extinctions have been widely reported across its distribution (Santos et al., 1994; Oliveira, 1995; Feijó et al., 2015; Neto and Alves, 2016). The current human-induced rarity due to continued habitat loss and intense hunting pressure led previous authors to treat this species as one of the most threatened mammals in Brazil (Feijó et al., 2015). The new populations reported in this study offer great opportunities to fill large knowledge shortfalls on the species biology and bring new opportunities for the conservation of the three-banded armadillo.

Nevertheless, most of the new records are close to those previously reported and located in similar habitats such as the uplands of central Caatinga and northeastern Cerrado. Therefore, the bulk of

**Table 1.** Temporal land cover composition (patch and core region) across the Brazilian three-banded armadillo distribution

| | Number of patches | | | | Total patch area (hectares) | | | |
|---|---|---|---|---|---|---|---|---|
| Classes | 1985 | 2000 | 2020 | Dif. (%) | 1985 | 2000 | 2020 | Dif. (%) |
| Savanna | 17,004 | 18,924 | 22,009 | 29.43 | 78,600,648 | 75,400,146 | 69,432,502 | −11.66 |
| Grassland | 24,288 | 24,392 | 25,234 | 3.89 | 13,703,821 | 12,986,367 | 11,749,838 | −14.26 |
| Forest | 31,559 | 28,898 | 31,802 | 0.77 | 12,723,765 | 11,316,981 | 11,584,369 | −8.95 |
| Pasture | 29,146 | 31,530 | 35,092 | 20.40 | 18,015,679 | 24,498,873 | 26,201,419 | 45.44 |
| Agriculture | 1,743 | 2,059 | 4,293 | 146.30 | 395,330 | 1,677,109 | 5,607,238 | 14,083.69 |
| Mosaic of agriculture and pasture | 47,746 | 42,270 | 46,296 | −3.04 | 11,665,550 | 9253777 | 10,341,375 | −11.35 |
| Others | 14,057 | 12,816 | 13,066 | −7.05 | 2,851,765 | 2,823,474 | 3,039,902 | 6.60 |
| | Number of cores | | | | Total core area (hectares) | | | |
| Classes | 1985 | 2000 | 2020 | Dif. (%) | 1985 | 2000 | 2020 | Dif (%) |
| Savanna | 7,432 | 7,095 | 6,675 | −10.19 | 15,438,353 | 13,316,214 | 10,086,497 | −34.67 |
| Grassland | 818 | 757 | 611 | −25.31 | 1,159,797 | 839,771 | 681,354 | −41.25 |
| Forest | 615 | 550 | 554 | −9.92 | 1,048,980 | 897,698 | 790,492 | −24.64 |
| Pasture | 998 | 1,752 | 1,838 | 84.17 | 433,947 | 1,060,062 | 1,199,422 | 176.40 |
| Agriculture | 11 | 138 | 437 | 3,872.73 | 3,106 | 183,854 | 1,155,851 | 37,113.49 |
| Mosaic of agriculture and pasture | 172 | 139 | 172 | 0.00 | 61,704 | 82,272 | 126,012 | 104.22 |
| Others | 39 | 72 | 84 | 115.38 | 245,895 | 230,868 | 251,016 | 2.08 |

*Note:* Dif. represents the percentage of patch change across classes between 1985 and 2020.

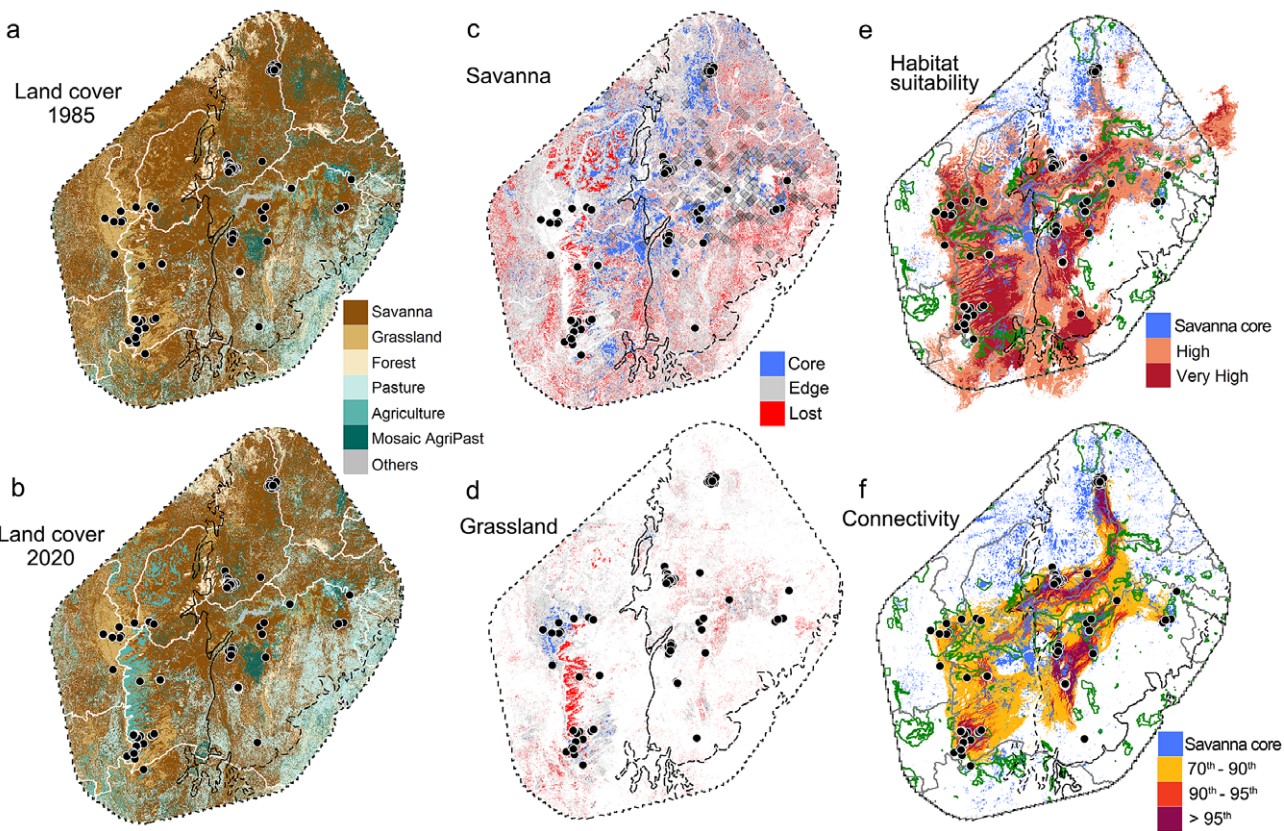

**Figure 3.** Temporal land cover change across the distribution of the Brazilian three-banded armadillo and priority areas for reactive and proactive conservation actions. *Note:* Left panels show land cover distribution in (a) 1985 and (b) 2020 based on MapBiomas collection 6. Maps show the six main land cover classes (representing 97% of the area). 'Others' include mangrove, wetlands, urban areas, dunes, and forest plantations. Middle panels show the difference in land cover (between 1985 and 2020) for patches of natural savannas (c) and grasslands (d). Red cells represent cells of natural land lost in the last 35 years. Blue and gray cells represent the core and edge areas, respectively, of the remnant natural habitat patches. Gray diamonds, in panel (c), represent historical records of the species. Right panels show existing protected areas (green polygons) and core savanna areas (blue cells) that overlap with distribution of the species (e), and the main corridors for connecting the remnant populations according to the Circuitscape connectivity model (f). In all panels, black dots represent recent records of the species. Black dashed lines delimit the biomes (according to Instituto Brasileiro de Geografia e Estatística (IBGE), 2004) and white lines delimit the Brazilian states. The region displayed represents a 200-km convex hull surrounding the marginal recent records.

historical records located in lowlands likely represent populations now extinct. Not surprisingly, historical records are concentrated in regions facing severe habitat loss, suggesting that the three-banded armadillo depends on natural core lands for survival. Accordingly, studying habitat selection of a congeneric species, Attias et al. (2018) found natural cover as an important characteristic of the landscape for the daily activities of the Southern three-banded armadillo (*Tolypeutes matacus*) in savannas and wetlands of mid-western Brazil. Thus, preserving the remnant core areas of savanna and grassland in the Caatinga and Cerrado should be a high priority to ensure the species' long-term survival.

Our landscape analysis reveals that 76% of natural core areas across the species range were converted by anthropogenic use in the past three decades. This alarming scenario suggests that previous estimates of the three-banded armadillo decline, where half of the original population was estimated to be lost (Reis et al., 2015), might be underrated. Our findings can thus improve the evaluation of the species conservation status according to IUCN criteria, which was last updated in 2013. Using natural core area loss as a proxy of population decline in the past 30 years (which is approximately the length of three generations for the species; Pacifici et al., 2013), we suggest that the Brazilian three-banded armadillo should be reclassified as Endangered based on criterion A2c, instead of its current Vulnerable IUCN status.

Comparing the populations of the two regions, those inhabiting Cerrado appear in a more critical situation. Most of the known Cerrado records are limited to preserved areas in the northwest and south portions of the species range, isolated from each other by a vast agricultural matrix. Such isolation brings additional risks of extinction related to demographic and environmental stochasticity and inbreeding depression (Hartley and Kunin, 2003; Benson et al., 2016). For example, Moraes (2015) found a very low genetic diversity in three-banded armadillos from a Cerrado population, which might suggest evidence of ongoing inbreeding depression. This shows that efforts should be directed not only at preserving populations on isolated patches within PAs but also at facilitating dispersal of individuals between populations in a way to maintain genetic diversity and guarantee their long-term viability (Benson et al., 2016; Thatte et al., 2018). Our connectivity models highlight important areas with a greater likelihood of dispersal that, if preserved, may facilitate gene flow among three-banded armadillo populations within and between biomes and ultimately increase their genetic diversity and survival likelihood. Many of these potential corridors are outside of existing PAs and embedded within a mosaic of anthropogenic land uses, being very likely to be degraded in the near future and demanding reactive actions, unless private landowners and managers are engaged in the conservation strategies for the species (Redpath et al., 2017).

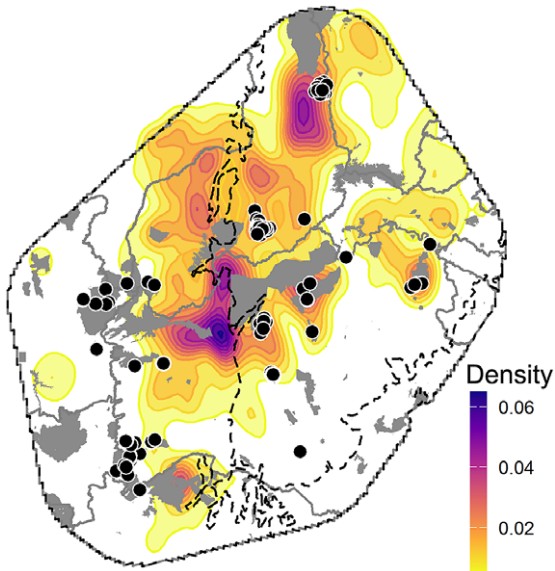

**Figure 4.** Key areas for proactive conservation actions. *Note:* Extant protected areas within the Brazilian three-banded armadillo range are shown in gray polygons and heat map reflects the density of remnant savanna core areas. Black dots represent recent three-banded armadillo records, black dashed lines delimit the biomes and gray lines delimit the Brazilian states. The region displayed comprehends the 200-km convex hull from the marginal recent records. Note that a large density of core savanna remnants are outside of extant PAs and they form a large corridor that, if preserved, could secure animal dispersal between Cerrado and Caatinga and benefit multiple threatened open-dweller taxa associated with savanna-grassland areas.

Most remnant core areas of savanna are located in the north-western portion of Caatinga and the Caatinga–Cerrado transition (Figure 4). A large part (84%) of these core areas are unprotected, highlighting the need to expand the PA network and implement other biodiversity-friendly management measures in the region (Leal et al., 2005; Fonseca et al., 2018). Caatinga is one of the least protected regions in Brazil with only 8% of its area under some type of legal protection and only 1.3% under strict protection from human use (Teixeira et al., 2021). Similarly, about 76% of the recent records of the Brazilian three-banded armadillo are located in unprotected private lands. Therefore, in addition to the implementation of public PAs, initiatives aiming at preserving and restoring native vegetation within private lands, such as the creation of private reserves (which receive fiscal incentives from the Brazilian government) must be encouraged.

Worrisomely, in the past years, Brazil has witnessed an alarming increase in deforestation rates due to weakened environmental legislation and surveillance (Vale et al., 2021). There have been steep reductions in funding for the creation of new public PAs and surveillance of the existing ones. Nevertheless, the Brazilian Forest Code requires landowners to preserve the natural vegetation on at least 20% of their properties. Therefore, these actors could play a key role in the enhancement of landscape connectivity outside PAs, just by complying with the current environmental legislation (Redpath et al., 2017). Landowners can also contribute to the species survival by adopting more sustainable land use practices adapted to the semiarid environment (Niemeyer and Vale, 2022), such as inhibiting illegal hunting in their properties, reducing or ceasing the clearing of native vegetation, and actively or passively restoring native vegetation to fulfill their legal conservation requirements. Undoubtedly, the conservation of the Brazilian three-banded armadillo will rely on the development of a healthy relationship with local stakeholders (Lees et al., 2021).

Based on the location of extant PAs and natural core remnants, we identify key areas for the establishment of new PAs or low-impact land management practices (Figure 4). These areas form a large corridor that could secure dispersal between Cerrado and Caatinga and benefit the biodiversity associated with savanna-grassland areas. For example, some emblematic species on the brink of extinction in northeastern Brazil, such as the jaguar, cougar, giant anteater, giant armadillo, brocket deer, and peccaries, are now mainly restricted to these core areas (see Supplementary Figure S3). On the other hand, the eastern portion of the Caatinga comprises most of the areas previously classified as of extremely high restoration importance (Fonseca et al., 2018), reflecting the highest rate of habitat loss. This scenario is supported by the fact that most of the documented three-banded armadillo local extinctions are located in eastern Caatinga. Hence, the few known remnant three-banded armadillo populations in this area require reactive conservation actions.

The three-banded armadillo can thus be viewed as a flagship species and draw attention to other threatened open-dweller taxa usually neglected in conservation planning. This role was recently strengthened when it was chosen as the mascot for the 2014' FIFA World Cup held in Brazil, attracting greater scientific, political, and social attention (Melo et al., 2014; Bernard and Melo, 2019). As a result, PAs were created, conservation and scientific projects focusing on the species were initiated, and the PAN Tatu-bola was published (Bernard and Melo, 2019). Additionally, three-banded armadillos are well valued among local people (Magalhães et al., 2022). Popularity and risk of extinction are important factors when defining flagship species and seeking engagement of the general public and local people for conservation initiatives (Bowen-Jones and Entwistle, 2002).

In addition to habitat loss, hunting is considered an important threat to the three-banded armadillo (Coimbra-Filho, 1972; Santos et al., 1994; Reis et al., 2015). The defensive strategy of the genus *Tolypeutes* is unique among armadillos as it relies on rolling up its body completely into a ball and remaining immobile (Figure 1). While efficient against natural predators, it makes three-banded armadillos extremely vulnerable to hunters (Santos et al., 1994). Ethnozoological studies recognized overhunting as one of the causes leading Brazilian three-banded armadillos to local extinction (Neto and Alves, 2016; da Silva Neto et al., 2017). This might explain the extirpation of populations from areas still holding sizable natural patches and the association of remnant populations with core areas of habitat patches, likely less accessible to hunters. Moreover, we expect that habitat loss and hunting act as interacting drivers of population decline, which greatly elevate the likelihood of extinction (Kimmel et al., 2022). For example, individuals are likely to be more exposed in agricultural lands and fragmented habitats, which is likely to increase hunting pressure (Romero-Muñoz et al., 2019). The establishment of PAs can alleviate habitat loss and hunting, but illegal hunting can still be common within PAs in Brazil (Ferreguetti et al., 2018; Dias et al., 2020). Nevertheless, PA establishment is only effective for resource conservation when accompanied with proper funding for infrastructure and personnel that can enforce protective measures within park boundaries (Terborgh and van Schaik, 2002). Educational campaigns and the engagement of local communities in conservation initiatives can also be effective tools to change pro-environmental behavior of local stakeholders and thus reduce hunting pressure (Cureg et al., 2016). Therefore, a socio-economic evaluation of the areas of conservation priority should be performed to plan the most effective conservation actions. This will enable the understanding of local

human-wildlife coexistence dynamics, shedding light on matters such as hunting motivations and rates, and local people's perceptions toward the species and wildlife in general (Ban et al., 2013; Ferraz et al., 2022).

## Conclusions

Designing conservation actions for poorly known species is challenging. Here, we showed that valuable information for more efficient conservation strategies can be derived using biogeographical analyses combined with landscape assessments documenting temporal land conversion. Our integrative approach uncovered key habitat and ecological conditions dictating species distribution, highlighted main corridors for connecting remnant populations, and identified priority areas for reactive and proactive actions. This set of information are preconditions for successful long-term conservation and is complementary to cost-effectively guide research and resource allocation. Importantly, they can be readily replicated for other poorly known species of which only limited presence-only occurrence records are available.

An important finding of our study is that the few known remnant populations of the Brazilian three-banded armadillo are closely associated with core areas of savanna and grassland in Caatinga and Cerrado. Preserving these natural core lands should thus be seen as a high priority in future conservation actions. Efforts should also be directed to enhance connectivity among the populations to guarantee their long-term viability. Based on the extant PAs and available core natural lands, we identify key areas forming a large corridor linking most of the remnant populations of the three-banded armadillo. This corridor represents a great opportunity for proactive conservation practices that will benefit multiple threatened species across the Cerrado and Caatinga, but conservation planning in these areas should also consider socio-economic factors and engage private stakeholders to make it feasible and long-lasting.

**Open peer review.** To view the open peer review materials for this article, please visit http://doi.org/10.1017/ext.2022.2.

**Supplementary materials.** To view supplementary material for this article, please visit http://doi.org/10.1017/ext.2022.2.

**Data availability statement.** Recent occurrence records of the Brazilian three-banded armadillo are available in Supplementary Table S1 and historical records were retrieved from Feijó et al. (2015). Land cover data used in this study were obtained from Mapbiomas collection 6 and are available at https://mapbiomas.org/download.

**Acknowledgments.** We are grateful to Diego Cavalheri, Jefferson Mikalauskas, Natan Freitas, Rodrigo Massara, Sarah Mangia, Wylde Vieira, and the members and collaborators of PAN Tatu-bola and PAN TATA for sharing occurrence records of three-banded armadillo and for providing valuable additional information.

**Author contributions.** Conceptualization: A.F., R.M.A., N.A.; Data curation: A.F., R.M.A., A.B., L.M.M.S.; Formal analysis: A.F.; Methodology: A.F., J.L.P.C.; Writing—original draft: A.F., R.M.A., N.A.; Writing—review and editing: A.F., R.M.A., A.B., J.L.P.C., L.M.M.S., N.A.

**Financial support.** This work was supported by the Second Tibetan Plateau Scientific Expedition and Research Program (A.F., Grant Numbers 2019QZKK0402 and 2019QZKK0501), the Chinese Academy of Sciences President's International Fellowship Initiative (A.F., Grant Number 2021PB0021), the Arizona Center for Nature Conservation (ACNC)/Phoenix Zoo Conservation & Science Grant Program (R.A.M.), the Coordenação de Aperfeiçoamento de Pessoal de Nível Superior (R.A.M., Grant Number 88887.199565/2018-00; L.M.M.S., Grant Number 330 88882.184251/2018-01), and the CNPq (N.A., Grant Number 141189/2007-0).

**Competing interest.** The authors declare none.

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
