## [Reviewer Report]

*Comments to Author*: Manuscript ID EXT-22-0039: Defining priority areas for proactive and reactive conservation practices for poorly known species: a case study of the endemic Brazilian three-banded armadillo

The authors combine models of species distribution and connectivity with temporal land cover changes to stablish priority areas for conservation of the endemic Brazilian three-banded armadillo (Tolypeutes tricinctus), one endemic and threatened species. They showed that the few remnant populations are spatially associates with core areas of savanna and grassland, which has been lost for agriculture in the past 30 years. The authors indicate key areas for proactive and reactive conservation actions for the three-banded armadillo, and highlight that these actions will benefit other threatened sympatric species.

Minor reviews

Line 43 – Include Tolypeutes as a keyword.

Line 121 – “The Caatinga is a biome…” The definition of "Biomes" provided by the Brazilian IBGE agency differs from what ecologist all over the world understand as Biomes. Therefore, it is proper use the term "Ecoregion" here, as well as in lines 310, 344, 346, 359, 367, 372, 433, 440, 706, 713, 725, 740, 746 and 750. Alternatively, the authors must explicitly in Methods that they are using the "Biomes" definition provided by the Brazilian IBGE agency, and include a reference.

Line 165 – “30° sec arc” Seems to have something wrong here. Is it a 30 degrees arc or a 30 seconds arc?

Line 189 - Query: Why the PC1 and PC2 axes range from -10 to 5 in Fig. 2b, and from -1 to 1 in Fig. 2C?

Line 192 – “…30’ sec arc…” It is unclear. Is the symbol ' means minute? Is the abbreviation "sec" means seconds?

Line 261 – Please, use bold type in “Results”

Line 262 – Please, do not use bold in this sub-title.

Lines 282-283 - Figure 2 has too many panels. In addition, the amount of information transmitted by Figs. 2d and 2e are not so impressive when compared with the other panels. Therefore, Figs. 2d and 2e should be excluded and the information included in there can be expressed in tables in Supp. Mat.

Line 293 – Figure 3 also has too many panels. Because most of the information of Figs 3a and 3b are already in table 1, I suggest excluded them from Fig 3. Figs 3a and 3b could appears in the Supp. Mat.

Line 308 – “Overlapping the mapping…” Consider rephrase

Line 395 – “Its defensive strategy is unique among armadillos…” Authors should clarify that it is a characteristic of the genus Tolypeutes, to be fair with the T. matacus.

Lines 710-726: The caption should be revised if Fig. 2d and Fig. 2e are excluded or moved to the Supp. Mat.

Lines 727-742: The caption should be revised if Fig. 3a and Fig. 3b are excluded or moved to the Supp. Mat.

---

## [Reviewer Report]

*Comments to Author*: The manuscript brings an interesting analysis that combines a modelling of the potential distribution of an endangered armadillo species with the dynamics of loss of native vegetation in the last 30 years. The focal species is endemic to the Brazilian Caatinga, a xeric biome relatively little studied and heavily impacted by anthropic activities. The results show a large loss of native vegetation, savannah, and grasslands, within the distribution area of the species in the last 30 years. This period represents about 3 generations of the target species, if we assume that the values estimated in (Pacifici et al. 2013) are correct (the generation length estimated as 4287 days or approximately 11.7 years). I believe it is important to mention this because the conservation status of the species is outdated and one of the criteria for inclusion of species in the IUCN red list is based on the observed or estimated size of population loss within the last 10 years or three generations, whichever is longer (criterion A). The analysis made, quantifying the amount of suitable area lost in this period, can be a valid proxy of the equivalent population loss, thus contributing to the review of its conservation status. But, aside from that, my main suggestion for improvement concerns the need for a clearer approximation of the narrative to the Brazilian forest code, an important instrument for the protection and restoration of native vegetation within private rural properties in Brazil. According to the forest code, Brazilian rural landowners are obliged to protect native vegetation within their properties, either in the form of legal reserves (LR) (20% in legal reserves within each property located in the Cerrado or Caatinga), or in the form of permanent protection areas (APPs) (along springs, streams, rivers, slopes, hilltops among other places). I consider important to bring this topic to the manuscript since one of the main results indicates that most of the area of suitable habitat for the species is outside the boundaries of protected areas and therefore within private properties. Thus, landowners should be viewed as one of the most important stakeholders to be involved in the conservation of this species. They can and should be called upon to participate both with reactive measures, such as preventing illegal hunting within their properties or preventing the clearing of native vegetation, and in proactive measures involving, for example, implementation and restoration of APPs and RLs within their properties. The creation of private nature reserves (RPPNs) can and should be listed in the manuscript as another proactive measure based on rural landowners. In fact, the Discussion should be more specific in delineating what should be those reactive or proactive measures. In the current format, the text is a bit generic on this point, lacking specificity to the study region. Returning to my point, the fact that the target species is charismatic should make it easier to convince these stakeholders to become active members in the protection of the species. With this, however, I do not want to draw emphasis away from the creation of new public protected areas in these regions indicated as highly suitable. Pragmatically speaking, the recent history of Brazil shows that this process is usually lengthy and strongly dependent on political will, as is very well illustrated in the current anti-environment administration of the current president Bolsonaro. Another more specific points are indicated below.

Material and Methods

l. 126: “Recent estimation shows that 63% of the Caatinga has been converted into

anthropogenic ecosystems (Silva and Barbosa, 2017)”. Collection 6 of MapBiomas shows a very different figure for native vegetation loss in the Caatinga biome. According to this source, currently 34.9% and 64.04% are anthopogenic and natural cover, respectively. Please correct.

See data here: https://plataforma.brasil.mapbiomas.org/

l. 133: “Cerrado is estimated between 40-50% mainly caused by monocultures and pastures (Strassburg et al., 2017; Vieira et al., 2019).” See current figures in MapBiomas Collection 6 for the Cerrado biome: 53.9% (natural) and 45.9% (anthropic). Please correct.

Discussion

l. 360-onwards: “Unfortunately, many of these potential corridors are outside of existing protected areas and embedded within a mosaic of anthropogenic land uses, being very likely to be degraded in the near future and demanding reactive actions”. See above my comment on forest code and rural owners. Policies, incentives, and campaigns to Increase their adherence to the forest code could increase, in the long term, recovery of currently deforested areas of permanent preservation and legal reserves existing in rural properties.

Cited References

Pacifici, M., L. Santini, M. Di Marco, D. Baisero, L. Francucci, G. Grottolo Marasini, P. Visconti, and C. Rondinini. 2013. Generation length for mammals. Nature Conservation 5.

---

## [Editor Report]

*Comments to Author*: Many thanks for submitting what is an interesting study into a poorly researched species but also a poorly researched landscape. The manuscript in the main is well written and the methods are appropriate. One of the reviewers highlighted the importance of this research for Red List classification as well as the importance of private protected areas. I believe including these points and other minor points highlighted by both reviewers will strengthen the manuscript.

---

## [Reviewer Report]

*Comments to Author*: In my previous revision I made comments, requests for adjustments and improvements that were, in my opinion, duly addressed in this new version. Having nothing more to add, I congratulate the authors for their valuable contribution to the conservation of this threatened species of armadillo, endemic to Brazil.

---

## [Editor Report]

*Comments to Author*: Many thanks for considering the suggestions of the reviewers. The addition of the new paragraph in the discussion and the extension of the previous paragraph certainly strengthen the discussion and the paper. I am therefore happy, like the reviewers (many thanks to them for their time), to recommend the paper be accepted.